# Predicting COVID-19 Infections in Eswatini Using the Maximum Likelihood Estimation Method

**DOI:** 10.3390/ijerph19159171

**Published:** 2022-07-27

**Authors:** Sabelo Nick Dlamini, Wisdom Mdumiseni Dlamini, Ibrahima Socé Fall

**Affiliations:** 1Department of Geography, University of Eswatini, Kwaluseni, Manzini M200, Eswatini; wdlamini@uniswa.sz; 2World Health Organization, 1211 Geneva, Switzerland; socef@who.int

**Keywords:** COVID-19, Eswatini, risk mapping, Poisson regression

## Abstract

COVID-19 country spikes have been reported at varying temporal scales as a result of differences in the disease-driving factors. Factors affecting case load and mortality rates have varied between countries and regions. We investigated the association between socio-economic, weather, demographic and health variables with the reported cases of COVID-19 in Eswatini using the maximum likelihood estimation method for count data. A generalized Poisson regression (GPR) model was fitted with the data comprising 15 covariates to predict COVID-19 risk in the whole of Eswatini. The results show that the variables that were key determinants in the spread of the disease were those that included the proportion of elderly above 55 years at 98% (95% CI: 97–99%) and the proportion of youth below the age of 35 years at 8% (95% CI: 1.7–38%) with a pseudo R-square of 0.72. However, in the early phase of the virus when cases were fewer, results from the Poisson regression showed that household size, household density and poverty index were associated with reported COVID-19 cases in the country. We then produced a disease-risk map of predicted COVID-19 in Eswatini using variables that were selected by the regression model at a 5% significance level. The map could be used by the country to plan and prioritize health interventions against COVID-19. The identified areas of high risk may be further investigated to find out the risk amplifiers and assess what could be done to prevent them.

## 1. Introduction

COVID-19 has spread dramatically since it was first discovered in China in December 2019 [1]. In as much as the spread resulted in a global pandemic [2], countries continued to report national spikes of infections at different temporal scales as the dominant strain of the virus took its toll on the population. Studies had reported various determinants of country COVID-19 spikes and they included factors such as population density, adherence to preventive measures and a host of typical socio-economic factors [3]. The disease spreads through direct and effective contact with infected persons; as a result, mechanical barriers limiting population interactions and mixing, such as social distancing, had been enforced by many countries [4]. Understanding the environmental risk factors, as well as the socio-economic and demographic factors associated with the spread of COVID-19, is crucial for effective prevention and response efforts [5]. Often, the underlying factors would not always be exactly the same between affected countries as population behavior, structure and socio-economic activities may vary from country to country [6]. For instance, the onset and severity of the disease and its associated country spike had often been seen by disproportionate case loads and mortality numbers between countries [7]. The factors affecting the spread of the pandemic, case load and mortality rates had been studied at national and subnational levels and the identified key drivers varied between countries and regions. A study on COVID-19 age mortality curves had shown that the spread of COVID-19 was different for high-income countries (HIMs) and low–middle-income countries (LMICs) [8]. Interestingly, another study by [9] found that there were different susceptibilities and vulnerabilities to COVID-19 in urban and rural populations in the United States. These findings showed that the geographic context was a key determinant in the spread and severity of the disease.

Eswatini (formerly known by its English name as Swaziland), a country in the southeastern Africa region with a population of about 1.1 million, had not been spared by the pandemic. According to the Ministry of Health, this lower-middle-income country had reported cumulative cases of about 69,000 and total deaths of about 1400 people by February 2022. The government of Eswatini had also implemented strict lockdowns and travel restrictions during the peaks of the pandemic in 2020 and 2021 [10]. However, cases continued to soar in the midst of these restrictions and, in this study, we were interested in determining the factors that were the potential drivers of infections during the peaks of the disease waves and surges in Eswatini. For instance, Eswatini is known to have high poverty proportions at 58.9%, where 20% is considered to be extremely poor, high HIV infections at 26% and high tuberculosis rates, which could have increased susceptibility to the disease because of co-morbidities and other demographic factors [11].

Previous COVID-19 studies on the disease predictors and case-load forecasting included a host of clinical, epidemiological, demographic, and socio-economic factors [12,13,14]. In this study, we explored the association between various socio-economic, weather, demographic and health variables with the distribution and spread of COVID-19 cases in Eswatini using the maximum likelihood estimation method for count data. Count data are dispersed data that involve discrete variables and require discrete analysis methods to estimate the parameters. We then mapped and predicted the risk of COVID-19 for the whole of Eswatini using those variables that were significantly associated with the disease. We believe that the mapping products produced will not only help in the ongoing health intervention efforts against COVID-19 but they will also help in the identification of the risk amplifiers of the disease in the affected areas. This work is a continuation from our previous work, which used a similar set of predictors to perform a spatial risk assessment and identified epidemic risk prone areas based on susceptibility risk, transmission risk and exposure risk [15].

## 2. Materials and Methods

### 2.1. Study Area and Data Sources

Eswatini is a southern African country bordered by South Africa all around except on the northeastern side where it is bordered by Mozambique. COVID-19 had spread throughout the entire country and its severity varied between the four districts comprised of Hhohho, Manzini, Shiselweni and Lubombo. Figure 1 shows the geographic location and distribution of reported COVID-19 cases in Eswatini while Figure 2 shows the incidence of cases per 1000 population. As can be seen in the maps, COVID-19 had spread throughout the rest of the country since the first case was introduced in March 2020, affecting both rural and urban populations, but its risk severity varied from locality to locality. In this study, we were interested in investigating the determinants of this risk variation by locality. The country is landlocked and travel within and outside Eswatini is mostly by road. Informal crossings (mainly to South Africa and Mozambique) are also common along the borders, especially for the local communities residing along the international boundaries.

To map and predict the distribution of COVID-19 in the country, we used data ranging from weather, communication, internet connectivity, traffic density, health and demographic, as well as socio-economic variables. The data were made up of a list of 15 covariates, which were used as predictors and regressed with georeferenced case data via a generalized Poisson regression (GPR) model [16]. Data variables on location entities such as supermarkets, shopping and church distances were extracted from Google Earth. The health, demographic and socio-economic data used in this study were obtained from the central statistics office (CSO). The CSO data included household density, household size, human immunodeficiency virus (HIV) prevalence, population density, youth proportion and proportion of elderly above 55 years. Details for some of the data variables used in this study had been provided in our previous work [15]. COVID-19 case data comprising of reported cases ranging from the first case in March 2020 until December 2021 were obtained from the Ministry of Health [17]. The COVID-19 case data were made up of variables such as cases sex, age, case locality, clinical severity and date of hospital presentation. The data totaled 12,986 individual COVID-19 cases. The data were summed and aggregated at enumeration area (EA) level (made up of a total of 2326 EAs). The EAs serve as the smallest census units in the country, ranging from an area of about 0.013 km^2^ to about 194 km^2^. The data used in this study are described and presented in Table 1.

### 2.2. Statistical Modeling

In the first stage of the model, we conducted a bivariate stepwise regression to select the set of variables that were associated with COVID-19 at a 5% significance level and then we fitted the selected variables into the GPR model to estimate the incidence rate ratio (IRR) of the virus in Eswatini. The stepwise regression method allows the user to evaluate variables according to their order of importance in explaining the outcome variable of interest [18]. This way, we were able to select useful or significant subsets of the regressed variables in addition to identifying any variable enhancement or suppression [19] by adding or removing different single variables or a combination of variables in the model. In the final stage we used the variables selected in the second stage of the model to predict COVID-19 cases in the entire country using the maximum likelihood estimation method for count data. Association of COVID-19 symptoms with age was also assessed by categorizing individual ages into different groups to assess the most affected patients by age. Before selecting the model, we first investigated which model would be suitable for fitting the data between two more-or-less similar models involving the Poisson model and the negative binomial model [20] by assessing the presence of over-dispersion in the data. We checked for excess variation in the model by assessing whether the deviance of the model was greater than its corresponding degree of freedom. We also wanted to find out if any excess variation was due to over-dispersion or if it was due to excess zeros that may emanate from zero-reported cases in some of the EAs [21]. Since the negative binomial model is suitable in cases where there is over-dispersion or excess zeros, in our case we proceeded with the Poisson model as the deviance was not greater than its degree of freedom. We then developed a Poisson regression model that was first fitted with all 15 covariates using STATA statistical software version 13 [22] to select the most parsimonious set of variables that were associated with COVID-19. Variables that were selected at a 5% significance level were then added into the GPR model to predict the risk of COVID-19 in the whole of Eswatini. The Poisson model has only one parameter, *µ* (mean), which corresponds to the mean of the case counts. In our case, the counts (number of events) were weighted by the context in which they were occurring. In our study, we used the population in each EA as the denominator and the case counts that correspond to rates as the numerator, as shown below:
$$R=\frac{N}{P},\;\text{the}\;\text{Poisson}\;\text{regression}\;\text{then}\;\text{takes}\;\text{the}\;\text{form},\;\log\;(R_{i})=a+\beta X_{i},$$

which is equivalent to:
(1)
$$\log\;\left(N_{i}\right)=\log\left(P_{i}\right)+a+\beta X_{i}$$

where 
$\left(N_{i}\right)$
 is the rate of count in each EA (
i
) and 
P
 is the total population in that EA, 
a
 is the intercept and 
$\beta X_{i}$
 are the regression coefficients. In the GPR model, we used the maximum likelihood method to estimate the outcome of the counts of COVID-19 cases as follows:

Let *µ_ij_* be the mean count of 
$Y_{ij}$
 events, which in this model corresponds to rate of COVID-19 cases on each EA *i* on a particular date *j* and 
$X_{i}$
*~P (µ_i_,*
_1 −_
*µ_i_)* with likelihood:
(2)
$$P\left(Y_{ij};\mu_{ij}\right)={\frac{\mu_{ij}}{Y_{ij}!}}^{y_{ij}}e^{-\mu_{ij}}$$


Since the counts for *µ* can only be positive, we had to log transform the values so that the model can take both negative and positive values on either side of the equation. Therefore, when linearizing the model, we take the logarithm of µ as follows: 
(3)
$$\log(\mu_{i})=\log(N_{i})+a\;+\sum_{k=1}^{15}\beta X_{i}\;$$

where 
$N_{i}$
 is the total population in each EA used here as an offset to scale the modeling of rate 
$\mu_{i}$
 and 
βX
 are the regression coefficients corresponding to the 15 covariates. The Poisson regression, when applied on rates, assumes that the outcome is the count (numerator), while the log of the denominator, log(
$N_{i}$
), is considered as a covariate with regression coefficient fixed to 1 [23]. The maximum likelihood estimation method determines the values for the parameters in the model by maximizing the likelihood of observing the reported number of cases in each EA. The parameter values under investigation are found by maximizing the likelihood that the process described by the model produced the data that were actually observed in the original dataset. The model then maximizes the probability of obtaining the data we observed by estimating the mean number of cases and assuming that the counts of cases in the EAs were independent of each other. In our model, we assumed that the mean counts of COVID-19 cases in space and time followed a Poisson distribution, which is a discrete frequency distribution that gives the probability of a number of independent events occurring in a fixed space and time. The second model involved the fitting of the data into the same Poisson model but with only those variables that were selected at a 5% significance level in the first model fitting. The selected variables were used to predict the risk of COVID-19 in the whole of Eswatini using ArcGIS mapping software. COVID-19 predictions conducted in STATA were joined by EA using the ArcGIS spatial join tool. We predicted COVID-19 in unsampled locations in the country using the kriging method found in the ArcGIS spatial analyst tool. The number of interpolation points was set to 3 to force the model to estimate from the nearest sampled location and the search radius was set at 100 m to identify locality clusters associated with COVID-19.

## 3. Results and Discussion

### 3.1. Generalized Poisson Regression Model 

The results from the Poisson modeling with all 15 predictors showed that two variables were associated with COVID-19 at a 5% significance level. This was partly due to the rest of the socio-economic variables not varying in space when the virus became indiscriminate in its attacks and spread throughout the entire country. Therefore, variation by these variables did not hold and they became insignificant, especially when most of the EAs had reported at least one case. The selected variables included the proportion of elderly above 55 years of age at 98% (95% confidence interval (CI): 97–99%) and the proportion of youth below 35 years at 8% (95% CI: 1.7–38%) with a pseudo R-square of 0.72 (Table 2). These results were consistent with the findings by [24] on the clinical manifestations of the disease by age where they found that older people (age > 60) were at greater risk of developing severe symptoms from COVID-19 compared to younger people. Age remained an important mortality risk factor in a study by [25], while another study by [26] found that elderly people aged over 60 years exhibited more severe symptoms and had higher mortality rates among patients infected with COVID-19. Indeed, the difference in the proportion of elderly compared to the youth may also be explained by the fact that a greater percentage of the elderly were likely to present with clinical symptoms at health facilities and get tested, compared to the youth who were found to be mostly asymptomatic. Furthermore, studies have found that COVID-19 infections can be driven by asymptomatic young people [27,28].

In this study, a bivariate analysis of the clinical symptoms by age of COVID-19 patients showed that an average of 33-year-old (95% CI: 20–46 years) patients had no symptoms (Table 3). This age group was part of the proportion of youth and the working group in Eswatini. It also represents the most active group in terms of mobility, especially for livelihood and employment pursuits. A study by [29] found that the median age of COVID-19 patients with mild or moderate symptoms was 28 years. The findings in this study proved that a greater proportion of the youth who were infected with COVID-19 did not have any symptoms and thereby were less likely to get tested as the government testing strategy tended to be more reactive, especially in the earlier phase of the pandemic when testing kits were embarrassingly scarce. Furthermore, in the early stages of the pandemic, medical attention was given to the screening and management of symptomatic patients, and asymptomatic patients were only attended much later as clinical understating of the transmission of the virus improved.

Other variables such as church distance, shopping distance and supermarket distance were not significantly associated with COVID-19 infections and this may be attributed to the fact that most of these centers were closed and visiting such centers was severely restricted during the government’s strict lockdown measures [30]. Surprisingly, household size, household density and poverty index were not associated with COVID-19. However, other studies have found that household size and population density were associated with COVID-19 [31,32,33,34]. We wish to clarify that in the initial phase of the virus when cases were fewer and emerging, preliminary results from the Poisson regression showed that these variables were associated with COVID-19. Similarly, studies using early COVID-19 data until December 2020 mostly had the same findings [35]. In Eswatini, this was mostly because it was highly likely to find a positive case within the household of an index case as the Ministry of Health was conducting contact tracing following a positive case. Moreover, cases were spreading faster among people living within low-cost housing and poorer but congested communities who had little or no room to self-isolate in the country and this finding was similar to what [36] found in Africa and Latin America. However, as infections had spread country-wide, affecting both rich and poor communities, the association of these variables with COVID-19 had been confounded. Moreover, the data did not vary in space as most of the EAs reported at least a single case. Also, the information, technology and communication variables such as internet connectivity and cellphone usage were also not associated with COVID-19. Some studies have shown that reduction in cellphone activity at work and retail locations was associated with lesser growth in COVID-19 cases [37]. Clearly, the spread of the virus was indiscriminate as it attacked the rest of the country.

### 3.2. Age and COVID-19 Infections

The results of the bivariate analysis on the distribution of symptoms by age revealed that younger people aged <35 years were less likely to present with symptoms. It is possible that a majority of infected young people did not get tested and may have recovered without being reported anywhere. Table 3 presents the results of the bivariate analysis between clinical symptoms severity and age of COVID-19 patients. Symptoms were categorized according to severity where: no symptoms referred to those who had a positive test but did not show any clinical manifestations of the disease; mild symptoms were those who had a positive test and were barely showing any symptoms and quickly returned to feeling normal; moderate to severe referred to individuals who had any of the virus signs and symptoms such as fever, cough and sore throat, among others; and severe symptoms referred to individuals who required emergency treatment and hospital admission. Severe symptoms included patients who reported with shortness of breath, feeling faint or passing out, and having severe chest pains. Although COVID-19 had severe clinical manifestations among an average of 29-year-old patients, it is worth mentioning that this was a very small percentage (0.68%). This finding could be partly attributed to the fact that the first wave of COVID-19 in Eswatini was spreading faster among the active youth who were working, in schools and in tertiary institutions in the country and often had severe clinical manifestations in some of them. There was also a small proportion (8.2%) of individuals whose symptoms were not immediately ascertained and they were categorized as “unknown” in the COVID-19 database.

Variables that were selected at 5% significance level were used in the GPR model to predict the risk of COVID-19 infections and Figure 3 shows the predicted risk of COVID-19 in the whole country. The risk of COVID-19 was found to be higher (180 cases per 1000 population) in the surrounding urban and peri-urban areas of main towns and industrial complexes of the country comprising Piggs Peak, Manzini, Mbabane, Nhlangano, Bhunya, Mankayane, Matsapha, Siphofaneni and Big Bend. This was followed by rural and agricultural towns and other main places at 110 cases per 1000 population and these comprised Tshaneni, Simunye, Ngwenya and Sidvokodvo. The remainder of the rural areas and other sparse settlements in the countryside had less than 34 cases per 1000 population. A study by [38] also found that urban centers of the global south were highly likely to be predisposed to global risks such as COVID-19 because of their vulnerability and exposure, exacerbated by the process of urbanization in those centers. In Eswatini, urban centers are the backbone of daily activities and daily commuting between rural and urban centers is not uncommon. Indeed, cases reported in the earlier stages of the virus were mostly concentrated around urban and peri-urban areas and the daily contact between such places with rural commuters may have fueled the spread of the various to the rest of the country. The remaining areas with lower or zero cases were mainly private farms and rangelands with low population density and no residential areas. 

The investigation and understanding of the disease determinant factors by this study are not only crucial for the management and containment of COVID-19 but they are also crucial for the future management of similar pandemics. This present study investigated a collection of potential factors in the spreading of an infectious disease and found that there were disparities by age in its attack rate, symptom manifestations and mortality rate. Therefore, the social and demographic structure of a country determines the rate of the virus spread and its severity in the population. In addition, we found that it is important and also possible to manage infectious diseases such as COVID-19 in the very early stages of the onset of the virus when the driving factors could easily be predictable. For instance poverty, household density and household size may determine the rate of the disease propagation from an index case and studies have shown that quick isolation and quarantine of infected persons is crucial [39]. The spatial prediction in this study supports health intervention efforts as areas of heightened transmission can further be investigated to aid readiness and preparedness plan strategies even for similar future viruses. Our map products have shown that the virus has a potential to spread faster in the main places where people live and interact, such as urban areas, compared to the rural areas that were characterized by sparse and dispersed settlements. The demographic factors investigated in this study, such as age, can also be used to plan, for instance, the ongoing vaccination program, whereby the right age category can be targeted and prioritized.

## 4. Conclusions

We implemented a maximum likelihood estimation method based on the GPR model to discover the set of socio-economic, weather, demographic and health variables that were associated with COVID-19 in Eswatini. We found that the proportion of elderly population above the age of 55 years and the proportion of youth population less than 35 years of age were highly associated with the disease. Clearly, the virus had severe symptom manifestations among the elderly, prompting them to seek medical attention and subsequently get tested for the disease. Therefore, the presence of a higher population of elderly was an indicator for risk to the virus as they were more likely to fall sick compared to the youth who were found to be mostly asymptomatic. In addition, we found that the higher the presence of youth within the population, the higher the spread of the virus even though the youth were mostly asymptomatic. Other future studies could investigate if the presence of asymptomatic people is not a precursor for the mutation of the virus, as we have seen with the evolving strains of the COVID-19 pandemic. The association of the disease with other socio-economic variables gets confounded as the virus indiscriminately spreads to the rest of the population in a country. While the first analysis of the disease conducted during the early stage of the pandemic when cases were fewer showed that it was associated with poverty, household size and household density; these associations did not hold as most of these variables were later not significant as cases spread. Caution must be taken when interpreting such results as they also depend on the stage of the virus and its level of spread in a country, and the confounding and interaction of variables must be investigated.

In this work, we mapped and predicted the risk of COVID-19 infections in Eswatini using socio-economic, weather, demographic and health variables. The mapping products produced in this work could be used by the country to plan and prioritize health interventions for similar diseases in the future. The areas of high risk may be further investigated to discover the risk amplifiers in those areas and to assess what could be done to prevent them. Our work contributes to the ongoing COVID-19 surveillance and response efforts in the country and to the rest of the global fight against the virus in areas with a similar setting. More research work is needed to investigate how some of the variables used in this study impact the effectiveness of control interventions and how they could be used to aid preparedness planning and readiness for future pandemics in the country and elsewhere.

## Figures and Tables

**Figure 1 ijerph-19-09171-f001:**
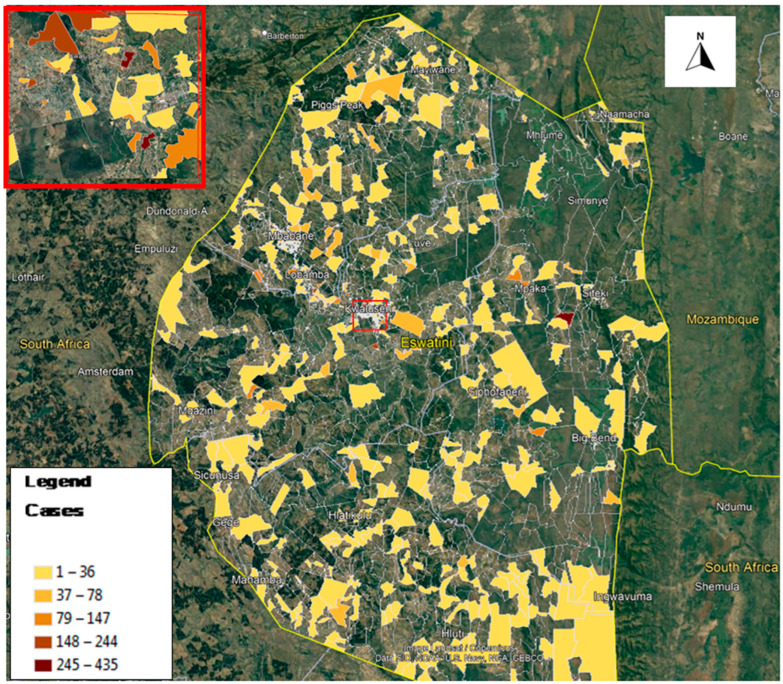
Distribution of reported COVID-19 cases in Eswatini.

**Figure 2 ijerph-19-09171-f002:**
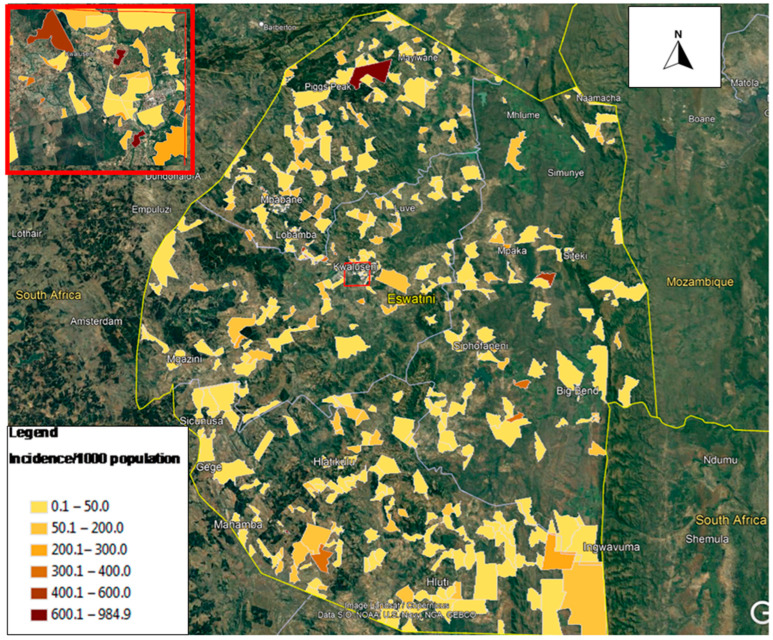
Distribution of COVID-19 cases per 1000 population in Eswatini.

**Figure 3 ijerph-19-09171-f003:**
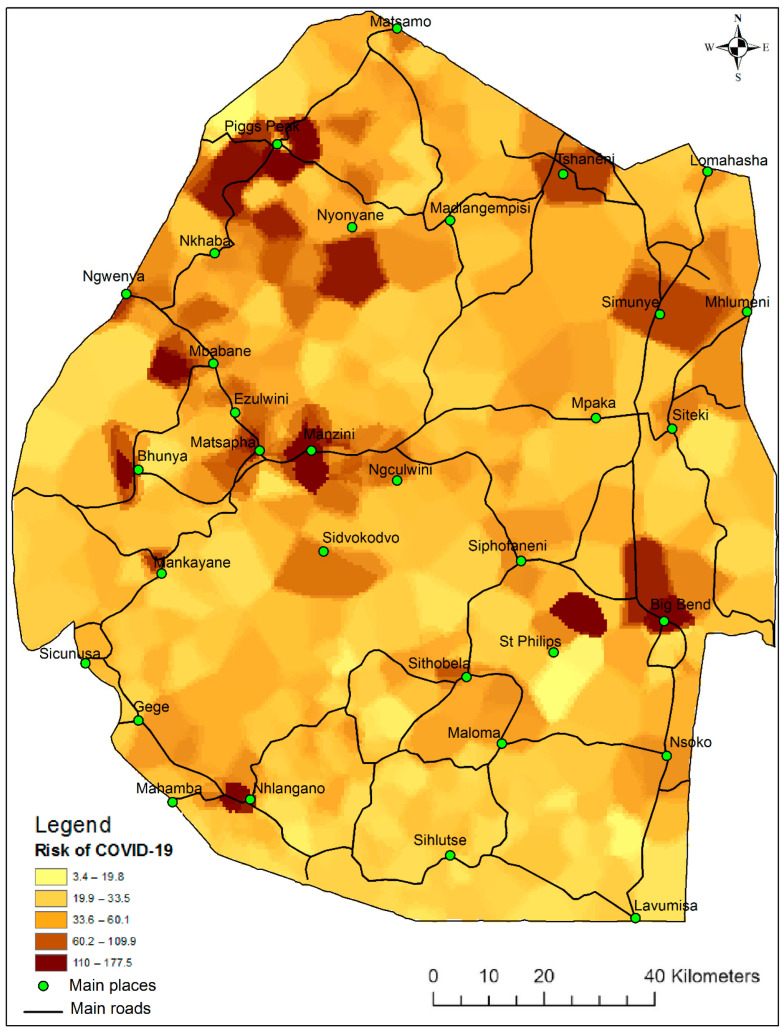
Predicted risk of COVID-19 infections in Eswatini.

**Table 1 ijerph-19-09171-t001:** Health, Demographic and Socio-Economic Data.

No.	Variable	Variable Short Name	Format	Description
1.	Cellphone usage	Cellphone	Floating point Values ranging from 0 to 1	Proportion of cellphone users per EA
2.	Church-distance	Church_dis	Floating point Distance (km)	Distance between EA and church
3.	Elderly above 55 years	Elderly_55	Floating point Values ranging from 0 to 100	Percentage or number of people above 55 years of age per 1000 people in each EA
4.	Household density	Hhld_dens	Floating point Number of buildings per unit area (km^2^)	Numerical quantities of the built up surface area in each EA
5.	Household size	Hhld_size	Integer number Number of people per household	Number of persons living in a private dwelling unit
6.	HIV Prevalence	HIV_prev	Floating point Values ranging from 0 to 100	Percentage of people living with HIV in each EA
7.	Internet connectivity	Internet	Floating point values ranging from 0 to 100	Percentage of people connected to internet either via a computer or other devices
8.	Poverty index	Po_index	Floating point Values ranging from 0 to 100	Percentage of people living below USD 2 per day in each EA.
9.	Building density	People_bld	Floating point Values ranging from 0 to 100	Percentage of built up area in each EA
10.	Youth proportion	Youth_prop	Floating point Values ranging from 0 to 100	Percentage or rate of people below 35 years per 1000 people of age in each EA.
11.	Shopping distance	Shop_dist	Floating point Distance (km)	Distance between EA and shopping area
12.	Supermarket distance	Supmkt_dis	Floating point Distance (km)	Distance between EA and supermarket
13.	Temperature	Temp	Floating point Degrees celcius	Hot/cold
14.	Traffic mean	Traff_mean	Floating point Number of vehicles moving through an area per day per unit area (km^2^)	Numerical quantities of average traffic moving through each EA approximated as a surface area of that EA
15.	Population density	Pop_dens	Floating point Number of people/per unit area (km^2^)	Numerical quantities of the populated surface area in each EA.
16.	Total population		Integer number Number of people	Number of people in the entire country obtained by summing up the number of people recorded in each EA

**Table 2 ijerph-19-09171-t002:** Results of the Poisson model.

Individual	IRR	Std Err.	z	P > z	95% CI	
cellphone	3.336945	2.693032	1.49	0.135	0.6861192	16.229
church_dis	0.991159	0.016569	−0.53	0.595	0.9592107	1.0242
elderly_55	0.984678	0.0025786	−5.9	0.000 *	0.979637	0.9897
hhld_dens	1.000045	0.0001841	0.25	0.806	0.9996845	1.0004
hhld_size	0.9565653	0.0408021	−1.04	0.298	0.8798463	1.04
hiv_prev	0.3712601	0.3929613	−0.94	0.349	0.046636	2.9555
internet	0.936553	0.4262686	−0.14	0.885	0.3838055	2.2854
p0_index	1.00156	0.0044225	0.35	0.724	0.99293	1.0103
people_bld	0.9836968	0.0358074	−0.45	0.652	0.9159606	1.0564
pop_dens	1.000002	0.0000865	0.03	0.977	0.999833	1.0002
youth_prop	0.0816543	0.0646422	−3.16	0.002 *	0.0173029	0.3853
shop_dist	0.9976175	0.0178658	−0.13	0.894	0.9632086	1.0333
supmkt_dis	1.002583	0.0159068	0.16	0.871	0.9718864	1.0343
temp	0.9545037	0.0276899	−1.61	0.108	0.9017466	1.0103
traff_mean	0.9999643	0.0000665	−0.54	0.591	0.9998339	1.0001

* Selected variables at 5% significance level; IRR = incident rate ratio, SD = standard deviation, z = Z score, P > z = significance-value or probability-value, 95% CI = 95% confidence interval for the estimated IRR.

**Table 3 ijerph-19-09171-t003:** Mean age by symptoms.

Mean Age by Symptoms				
Symptoms	Mean	SD	*N*	% *N*
No symptoms	33.4	13.75	5566	42.86
Mild	34.8	11.23	4681	36.05
Moderate to severe	48.0	16.97	177	1.36
Severe	29.0	12.35	88	0.68
Recovered	36.0	10.78	1236	9.52
Deceased	58.0	1.41	177	1.36
Unknown	36.2	9.39	1060	8.16
**Total**	**34.9**	**12.47**	**12,986**	**100**

## Data Availability

Supporting data from this study could be requested directly from the corresponding author following a reasonable request.

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
