# Peer review of "Predicting COVID-19 Infections in Eswatini Using the Maximum Likelihood Estimation Method"

_ijerph, 2022, doi:10.3390/ijerph19159171_

Round 1
Reviewer 1 Report
Although, the authors have now improved the presentation, there are still some technical issues that must be fixed before publication:
Table 2:
A precision of 4 digits after the dot is enough for all the reported values.
1) there are still problems with Table 2.1:
No. 5, household size: is this data an integer of a floating point value?
No. 7, the description is not conform to the format, does the number of people connected to internet ranges from 0 to 1?
No. 16. Total population: why this is a floating point value? shouldn't it be an integer?
Line 208: the value of the proportion of youth below 35 should be corrected, in the same way of line 16: 8% and 1.7%.
Table 3: at page 9: the term "recovered" usually means healed which is probably wrong and misleading. Does the authors mean admitted in hospitals? If this is the case is 36 years the correct average ?
Table 3: the year percentage of severe cases is 29 years this data is quite strange. Is this data correct? If yes the authors should give an interpretation of this strange result.
Table 3: considering the above comments, also to make the comparison with other studies more effective, the authors should explain the meaning of the different terms used for describing symptoms categories.
Table 3: the precision of standard deviation can be just 2 digits after the dot.
Author Response
Comments to Reviewer:
Although, the authors have now improved the presentation, there are still some technical issues that must be fixed before publication:
Table 2:
A precision of 4 digits after the dot is enough for all the reported values.
Response: Thank you for the comment, we have rounded all the values into 4 decimal places.
1) there are still problems with Table 2.1:
No. 5, household size: is this data an integer of a floating point value?
Response: We are very sorry for the missed change, household size is also an integer number and has been corrected in the table.
No. 7, the description is not conform to the format, does the number of people connected to internet ranges from 0 to 1?
Response: Our mistake, arising from juggling between probabilities and percentages. The values range from 0 to 100 and we have corrected the value range in the table. The values correspond to percentage of people connected to internet per enumeration area.
No. 16. Total population: why this is a floating point value? shouldn't it be an integer?
Response: Thank you for your comment, this was a mistake, we have corrected the value. It is an integer.
Line 208: the value of the proportion of youth below 35 should be corrected, in the same way of line 16: 8% and 1.7%.
Response: Thank you for the comment, we have corrected the values in line 208 to be similar to line 16.
Table 3: at page 9: the term "recovered" usually means healed which is probably wrong and misleading. Does the authors mean admitted in hospitals? If this is the case is 36 years the correct average?
Response: Thank you for your comment. Recovered represent those who were cured from COVID-19 and were no longer positive after a second PCR test. This usually took more than 30 days after showing symptoms. Patients were given medication and then told to self-isolate. If there were signs of improvement, the ministry of health personnel would then take a second test and if negative, would then declare the patient to have recovered from COVID-19. Yes, 36 years is the correct average.
Table 3: the year percentage of severe cases is 29 years this data is quite strange. Is this data correct? If yes the authors should give an interpretation of this strange result.
Response: Many thanks for the comment. As can be seen from the table, the number with severe symptoms was only 88 and this could be due to the fact that COVID infections were spreading faster among the youth and often had severe manifestations in some of them, which in actual fact is 0.68%. This was because during the first wave COVID-19 was spreading faster among the youth who were in schools and tertiary institutions in the country. We have added a sentence about this in line 283 to 288.
Table 3: considering the above comments, also to make the comparison with other studies more effective, the authors should explain the meaning of the different terms used for describing symptoms categories.
Response: many thanks for your comment, we have elaborated and clarified what is meant by the symptoms category in line 275 to 283 and 288 to 290.
Table 3: the precision of standard deviation can be just 2 digits after the dot.
Response: Thank you for your comment, we have edited the standard deviation of Table 3 to 2 decimal places.
This manuscript is a resubmission of an earlier submission. The following is a list of the peer review reports and author responses from that submission.
Round 1
Reviewer 1 Report
This paper presents an interesting study that aims to analyze the correlation between the spread of COVID-19 in Estwatini near South Africa with respect to several social and economical parameters.
However, there are aspects that should be clarified before considering it for publication:
The main problem is that the discussion is poor and not convincing, it seems that among the 16 selected features only two are significant, and the results are not surprising. Is figure 3 generated using these two features only? What is the role of the other? It would be possible to generate a risk map considering all the studied parameters?
The presentation should be improved as follows:
Section 2.1: Table 1:
- just three column should be enough Short name, format and description.
- the format column should describe the format of the data (e.g. types and ranges of values) using a kind of formal notation. Now it is similar to the description.
- Rows 1 and 2 are not described are they boolean types?
Section 2.2: the used method should be described better , more references and an example should be added.
Sections 3: more consideration about the other parameters should be discussed.
Minor points:
line 15: is this confidence interval correct?
line 51: the peaks of the pandemic
line 63: and requires discrete analysis methods to estimate the parameters.
line 134: please present the formula in a correct form.
Table 2: why cellphone is in bold?
Author Response
Response to Reviewer 1
Reviewer 1 Comments
This paper presents an interesting study that aims to analyze the correlation between the spread of COVID-19 in Estwatini near South Africa with respect to several social and economical parameters.
However, there are aspects that should be clarified before considering it for publication:
Comment 1:
The main problem is that the discussion is poor and not convincing, it seems that among the 16 selected features only two are significant, and the results are not surprising.
Answer to reviewer comment 1
Many thanks for your kind observations in our study. We have added more facts into the discussion section and also included more studies with similar findings. An explanation for the two selected variables had been added in addition to the one that was already made in the study in paragraph 2 of page 7.
Comment 2
Is figure 3 generated using these two features only? What is the role of the other? It would be possible to generate a risk map considering all the studied parameters?
Answer to reviewer comment 2
We thank the reviewer for this comment. The prediction is often based on only the variables that were significant because it means that they are the ones that explain better the observed association with the cases in the data. This means that the variables are the ones that generated the case data we have observed and therefore they are highly likely to predict the unobservable future cases. It is also possible to consider all studied parameters whereby their joint, marginal and conditional probability distribution may be used to predict the outcome of interest for instance using Bayesian methods.
Comment 3
The presentation should be improved as follows:
Section 2.1: Table 1:
- just three column should be enough Short name, format and description.
Thank you for this comment, we have kept the column long name because that is how the variables were presented in the text and explained. The short name was meant for modeling purposes and for the variables to be recognized when results were presented.
- the format column should describe the format of the data (e.g. types and ranges of values) using a kind of formal notation. Now it is similar to the description.
Thank you for your comment. We have edited the table columns on format and description and described the data format as suggested by the reviewer.
- Rows 1 and 2 are not described are they boolean types?
Thank you for the comment. We have added the correct description for row 1 and 2. They are not Boolean data type.
Section 2.2: the used method should be described better , more references and an example should be added.
Thank you for the comment. We have added more description and more references to the methods section.
Sections 3: more consideration about the other parameters should be discussed.
Thank you for your comment. A description of the other parameters was included before in line 172 to 179 of the first version where we clarified that these parameters were also significantly associated with the disease in the early pandemic stage. Other studies that have shown the importance of these variables had been added into this revised version. We had also made a strong point about them in the conclusion which was in line 226 to 233 in first draft. This point had been kept even in this revised version.
Minor points:
line 15: is this confidence interval correct?
Thank you for the comment. We have corrected in confidence interval in the revised manuscript.
line 51: the peaks of the pandemic
Thank you for the comment. We have made a corrections and changed ‘peak’ to ‘peaks’.
line 63: and requires discrete analysis methods to estimate the parameters.
Thank you for the comment. We have made the grammatical correction in the revised version.
line 134: please present the formula in a correct form.
Thank for the comment, the formula had been corrected.
Table 2: why cellphone is in bold?
Thank you for the comment. This was a mistake and it had been corrected in the revised version

Reviewer 2 Report
The authors proposed a paper to investigate the association between various socio-economic, demographic and health variables with the spread of COVID-19 cases in Eswatini using the maximum likelihood estimation method for count data. They also produced a risk map of predicted COVID-19 in Eswatini using the variables that were selected at 5% significance level, which could be used to plan and prioritize health interventions against COVID-19.
The paper contains a topic worth investigating. However, I would like to see the author's responses to the following points before considering the paper for acceptance:
1) It is not clear how the authors used the MLE algorithm.
2) How would the authors explain the high risk near the Big Bend area? We can see a small area with many cases on the maps, but it is an isolated area.
3) section 3.2 is confusing, and it is unclear how the data was produced.
4) The paper needs an English revision by a native.
5) The math expressions need to be better presented. The font and size have to be uniform throughout the text.
Author Response
Reviewer 2
The authors proposed a paper to investigate the association between various socio-economic, demographic and health variables with the spread of COVID-19 cases in Eswatini using the maximum likelihood estimation method for count data. They also produced a risk map of predicted COVID-19 in Eswatini using the variables that were selected at 5% significance level, which could be used to plan and prioritize health interventions against COVID-19.
The paper contains a topic worth investigating. However, I would like to see the author's responses to the following points before considering the paper for acceptance:
Thank you for your kind comment.
1) It is not clear how the authors used the MLE algorithm.
Thank you for this comment, we have added additional information on the MLE method and also included more references on the algorithm in this revised version.
2) How would the authors explain the high risk near the Big Bend area? We can see a small area with many cases on the maps, but it is an isolated area.
Thank you for your comment, these are places where there are high concentrations of settlements i.e main towns and smaller rural towns. Big Bend for instance, is an agricultural town, surrounded by large sugar cane fields and population density is only high closer to the town center. The rest of the areas with lower or no cases are mostly private farms and rangelands with no residential areas. This had been explained in the discussion section in the text.
3) section 3.2 is confusing, and it is unclear how the data was produced.
Thank you for your comment, we have corrected the data description and processing information and also explained the variables that were used in the model.
4) The paper needs an English revision by a native.
Thank you for your comment, the English and grammar had been revised throughout the paper.
5) The math expressions need to be better presented. The font and size have to be uniform throughout the text.
Thank you for your comment the formula had been corrected in the revised version.

Round 2
Reviewer 1 Report
There are still relevant problems in the presentation, apparently the paper was not proofread and the comment were addressed too quickly and in a naive way.
I am quite disappointed reading this revised version, it seems that authors have submitted their revised version quickly claiming the they addressed all the issues including proofreading by a native English speaker, without doing this.
Example line 109: " the data.....IS......" and many other.
Moreover at line 95 authors mention 15 variables used in the study, in the table variables become 16 and successively, line 151, they were reduced to 14. The information in the table is still non complete, descriptions of variables are missing.
Most, importantly the authors did not clarify the main points in particular, the CI interval of the proportion of youth was corrected without adding significant comments and explanation to this variable and CI. Moreover confusion is added in section 3.2. For example, line 218 mention "age > 35"?
The authors should take the revision process more seriously when submitting to an international Journal.
You could also consider to ask for a further revision.
Author Response
Response to Reviewer 1 Round 2 comments
There are still relevant problems in the presentation, apparently the paper was not proofread and the comment were addressed too quickly and in a naive way.
I am quite disappointed reading this revised version, it seems that authors have submitted their revised version quickly claiming the they addressed all the issues including proofreading by a native English speaker, without doing this.
We are very sorry about this. We have re-revised and re-proofread the whole manuscript. We have also made an effort to improve on the consistency of the facts and figures presented.
Example line 109: " the data.....IS......" and many other.
This was an oversight and we have corrected the error.
Moreover at line 95 authors mention 15 variables used in the study, in the table variables become 16 and successively, line 151, they were reduced to 14.
We are sorry about this. It was due to the fact that one variable (Population) was added as an offset in the model and we did not count it as a covariate. There were fifteen covariates and this error had also been corrected for the model formulation.
The information in the table is still non complete, descriptions of variables are missing.
We are sorry, this was missed by mistake. We have completed the information on Table 2.1
Most, importantly the authors did not clarify the main points in particular, the CI interval of the proportion of youth was corrected without adding significant comments and explanation to this variable and CI. Moreover confusion is added in section 3.2. For example, line 218 mention "age > 35"?
This was a mistake, we have corrected the symbol. Also, this finding was explained in the clinical manifestation of the disease in line 177 to 181 and following the bivariate analysis in line 217.
The authors should take the revision process more seriously when submitting to an international Journal.
You could also consider to ask for a further revision.
We thank you for the comment. We have made an effort to re-revise the manuscript and rechecked for consistency in the facts presented.

Reviewer 2 Report
The authors improved the manuscript.
Author Response
Thank you for your comments. We have re-revised the latest version.
Round 3
Reviewer 1 Report
This paper still has major presentation problems.
The authors should contact a serious native English professional able to understand and correct technical contents for this.
The Table should be corrected as follows:
Table 2.1, the column format should describe datatype and ranging information for example:
Cellphone Usage: Floating-point ranging from 0 to 1.
Church distance: Floating-point (km)
Elderly above 55 years: Floating-point ranging from 0 to 100
and so on for all the variables......
Moreover there still a lot of problems in the text everywhere, please do not send this paper back before doing this.